# In situ hydromechanical responses during well drilling recorded by fiber-optic distributed strain sensing

Yi Zhang[1,2], Xinglin Lei[3], Tsutomu Hashimoto[1,2], Ziqiu Xue[1,2]

[1] Geological Carbon Dioxide Storage Technology Research Association, Kyoto, 6190292 Japan.
   [2] Research Institute of Innovative Technology for the Earth (RITE), Kyoto, 619-0292 Japan.

   [3] Geological Survey of Japan, National Institute of Advanced Industrial Science and Technology, Tsukuba, 305-8567, Japan.

*Correspondence to*: Yi Zhang (zhangyi@rite.or.jp)

**Abstract.** Drilling fluid infiltration during well drilling may induce pore pressure and strain perturbations in neighboured reservoir formations. In this study, we report that such small strain changes (~20 με) have been in situ monitored using fiber-optic distributed strain sensing (DSS) in two observation wells with different distances (approximately 3 m and 9 m) from the new drilled wellbore in a shallow water aquifer. The results show the layered pattern of the drilling-induced hydromechanical deformation. The pattern could be indicative of (1) fluid pressure diffusion through each zone with distinct permeabilities or

(2) the heterogeneous formation damage caused by the mud filter cakes during the drilling. A coupled hydromechanical model is used to interpret the two possibilities. The DSS method could be deployed in similar applications such as geophysical well testing with fluid injection (or extraction) and in studying reservoir fluid flow behaviour with hydromechanical responses. The DSS method would be useful for understanding reservoir pressure communication, determining the zones for fluid productions or injection (e.g., for $CO_2$ storage), and optimizing reservoir management and utilization.


## 1 Introduction

The utilization of underground reservoirs includes the exploitation or storage of resources such as groundwater, oil/gas, heat, and more recently, the $CO_2$ for mitigating the effect of $CO_2$ emission on global warming (Benson et al., 2005), as well as storage of compressed air for electric energy storage (Mouli-Castillo et al., 2019) in underground reservoirs. For better

utilization, an understanding of fluid flow and reservoir characteristics is required for more manageable and optimized operations. Geophysical methods, such as site-scale seismic, electrical methods, and well logging, have been widely applied for reservoir characterization and monitoring.

Distributed fiber optic sensing is emerging as a novel and practical technology for underground reservoir monitoring by measuring the environmental changes of physical fields, such as temperature, strain, and elastic waves (Barrias et al., 2016;

Schenato, 2017; Shanafield et al., 2018). There have been numerous application studies using distributed temperature sensing (DTS) and distributed acoustic sensing (DAS) in subsurface monitoring. DTS data have been useful for understanding fluid flow behaviour (such as flow rate and active fluid flow zone) and reservoir characteristics owing to the hydro-thermal coupling in addition to heat transport monitoring (Bense et al., 2016; Freifeld et al., 2008; Luo et al., 2020; Maldaner et al., 2019; des Tombe et al., 2019). DAS has been intensively developed and used to monitor surface, subsurface shallow reservoirs or deep

structures (Daley et al., 2013; Jousset et al., 2018; Lellouch et al., 2019; Lindsey et al., 2019, 2020; Zhu and Stensrud, 2019). On the other hand, the usage of distributed strain sensing (DSS) for subsurface monitoring of quasi-static deformation is comparatively less.

Although the main purpose of DSS is the monitoring of geomechanical deformations or earth subsidence (for safety considerations) (Kogure and Okuda, 2018; Krietsch et al., 2018; Murdoch et al., 2020; Zhang et al., 2018), DSS could also be

used to understand reservoir formation and reservoir flow owing to hydro-mechanical coupling. In principle, the physical coupling between fluid flow and strain is understood by the linear poroelasticity theory (Biot, 1941). In poroelastic theory, the deformation, such as soil consolidation, can induce "solid-to-fluid" coupled pressure change and fluid flow, whereas conversely, the fluid flow with pressure change can modify the effective stress of reservoir formation and cause "fluid-to-solid" coupled deformations (Cheng, 2016; Neuzil, 2003; Wang, 2017). The deformations could be the expression of fluid

flow behaviour in the reservoir and bear information regarding fluid flow and reservoir characteristics (such as permeability and compressibility) (Barbour and Wyatt, 2014; Schuite et al., 2015, 2017; Schweisinger et al., 2009; Zhang and Xue, 2019). By monitoring strain changes of an aquifer, fluid-to-solid coupling can characterize the hydraulic parameters in the reservoir formation.

Deformation-based reservoir monitoring methods have been recently applied to obtain the lateral permeability distribution (at

coarse scales) of underground reservoirs with surface deformation monitored by InSAR technique (Bohloli et al., 2018a; Vasco et al., 2008, 2010) and estimate the vertical compressibility with vertical deformation measured by well-based techniques (e.g., radioactive maker technique and extensometer stations) (Ferronato et al., 2003; Hisz et al., 2013; Murdoch et al., 2015). However, such vertical deformation monitoring tools are usually only available at limited points and over limited time intervals. In addition, it is not well understood the contribution of each formation zone to the total surface displacement.

It could be suitable for in situ monitoring of such hydromechanical responses in reservoirs via the high accuracy and resolution of DSS using optical fibers. Several studies have used the DSS tool to demonstrate that the deformation recorded during fluid injection in rocks can be utilized to obtain information of permeability, compressibility, and track pressure and fluid plume migration in laboratory experiments (Zhang et al., 2019; Zhang and Xue, 2019). Becker et al. (2017), Lei et al. (2019) and Sun et al. (2020) have recent shown that the hydromechanical responses during reservoir testing (water injection or extraction)

can be effectively monitored via DSS. These studies suggest the high application potential of the DSS tool in field studies for monitoring underground fluid reservoirs.

In addition to the purposed reservoir testing, the well drilling process itself also develops hydromechnical processes—the drilling fluid (also called mud) can infiltrate the reservoir formation under the pressure drive from the wellbore and deform the

formation. Considering the hydromechanical response, the spatial variations in reservoir permeability heterogeneity are expected to affect the pattern of formation deformation. Conversely, the deformation pattern could be indicative of the formation permeability structure. Besides, the formation damage may be involved in the drilling process and affect the pressure diffusion process. The formation of mud filter cake near the wall of borehole and the infiltration of solid particles in drilling fluid may occur during the drilling and reduce the permeability around the borehole. This may also affect the hydromechanical deformation.

In this study, we examine the high-resolution DSS records of a field study with strain monitoring in two wells (where optical fiber cables were installed) while drilling a new well. The results suggest that the formation strain pattern during well drilling could be associated to two causes: either by the permeability structure or drilling-induced formation damage (or their combination). For the former cause, the data can be used to understand the reservoir lithological changes and permeability structure. In this paper, we first introduce the measurement principle of high-resolution DSS based on Rayleigh scattering and field site operations with considering the installation method, then present the results of monitoring using DSS while well drilling, and finally we interpret the strain pattern using a coupled hydromechanical numerical model and discuss the two possibilities of the two causes. Some implications and potential applications are emphasized.

## 2 Methods

Optical fiber sensors work with the principle that the environmental effects, e.g., strain, temperature, can alter the phase, frequency, spectral content, and power of backscattered light propagated through an optical fiber. There are three types of scattering mechanisms—Raman, Brillouin, and Rayleigh scattering—used for measuring temperature or strain changes. In this study, we only consider the Rayleigh backscattering based method.

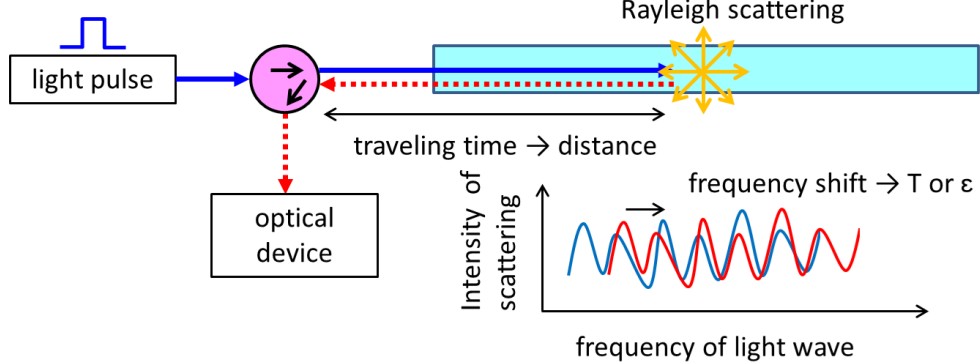

**Figure 1. Illustration of time-domain reflectometer (COTDR) method based on Rayleigh scattering.**

Rayleigh backscattering occurs when light propagates due to the existence of small random optical defects or impurities in the fiber core. Rayleigh backscatter spectrum of a point in an optical fiber can be considered as a fingerprint of the fiber. In

conventional coherent optical time-domain reflectometer (COTDR) method, Rayleigh backscatter spectrum generated for each region in the longitudinal direction of the optical fiber is obtained through measurement (Fig. 1) (Hartog, 2017). From the frequency shift between the reference Rayleigh-scattering power spectrum (RSPS) and a target RSPS using the cross-correlation method, the strain or temperature change at the point can be calculated. The distance of the scattering occurrence to the input end can be calculated using the travel time of scattered light. Because the length of light pulse in COTDR is large, the spatial resolution of conventional COTDR is low.

To obtain high spatial resolution, the pulse length of incident light must be shortened. However, if the pulse is shortened, the light pulse energy and thus the signal intensity of the backscatters are lowered, and the measurement accuracy becomes low at positions distant from the input end. For overcoming the limitations of conventional COTDRs, in the new tuneable wavelength coherent optical time-domain reflectometer (TW-COTDR) method, the tuneable wavelength distributed feedback laser and chirp signals by frequency sweeping and modulation methods are used to shorten laser light pulses while simultaneously ensuring sufficient pulse intensity (Kishida et al., 2014; Koyamada et al., 2009). To enhance the intensity of chirped signals and suppress the range side lobe, Gaussian amplitude modulation is performed. An inverse chirp filter is used to obtain RSPS in the analysis. Finally, the cross-correlation method is used for calculating the frequency shift amount of the spectrum, which is further used to calculate the strain or temperature change. TW-COTDR offers the ability of single-end accessing distributed measurements, high sensitivity, wide range of spatial resolutions, and measurements over long distances. Each distributed point (a short portion) along the entire length of an optical fiber can be taken as a sensing element.

The frequency shift ($\Delta f$) caused by strain and temperature changes ($\Delta \epsilon$ and $\Delta T$) can be linearly described using the following simple equation,

$$\Delta f = A\Delta\epsilon + B\Delta T \tag{1}$$

where $A$ and B (are the coefficients) relate the frequency shift to strain and temperature changes. Under the condition of constant temperature ($\Delta T = 0$, assumed in this study), the frequency shift ($\Delta f$) simply becomes proportional to the strain change ($\Delta\epsilon$) by A. A is –0.140 GHz/µε for the optical fiber used in this study. The value was obtained from a prior calibration measurement, which was conducted using the tensile tester with a displacement gauge. We used an optical interrogator NBX-SR7000 (Neubrex Co., Ltd., Japan) with TW-COTDR function in this study (Kishida et al., 2014). The instruments can provide high measurement accuracy (0.5 µε) and spatial resolution (5 cm), allowing for the monitoring of very small strains over long distances (~25 km) in a distributed manner.

## 3 Field study

The field test site is located in the rural area of Mobara city (Chiba, Japan). The subsurface formation of the site develops near-horizontal layered heterogeneity by the lithological changes of sandstone-mud alternations (Lei et al., 2019). There are two pre-existing vertical wells (obs1 and obs2) with prior installations of optical fiber cables, by installing optical fiber cables

behind the casing of the wellbore. In engineering practice, because the silica-fabricated nude optical fiber itself is thin and weak, the fabricated fiber cable using extrinsic reinforced jackets are necessary for protecting the central fiber core and practically installing the fiber in underground wellbores. A stainless steel wire reinforced cable (strain cable) was deployed. In the fiber cable, two stainless steel wires (SUS304 WBP) are assembled alongside the fiber core (SR15) in the polyolefin elastomer body (Fig. 3). During the installation, the cable with each segment of steel casing was carefully placed downward to the wellbore. The cable was fixed using specially designed clamps, placing the fiber cable between the casing and the formation (Fig. 2c).

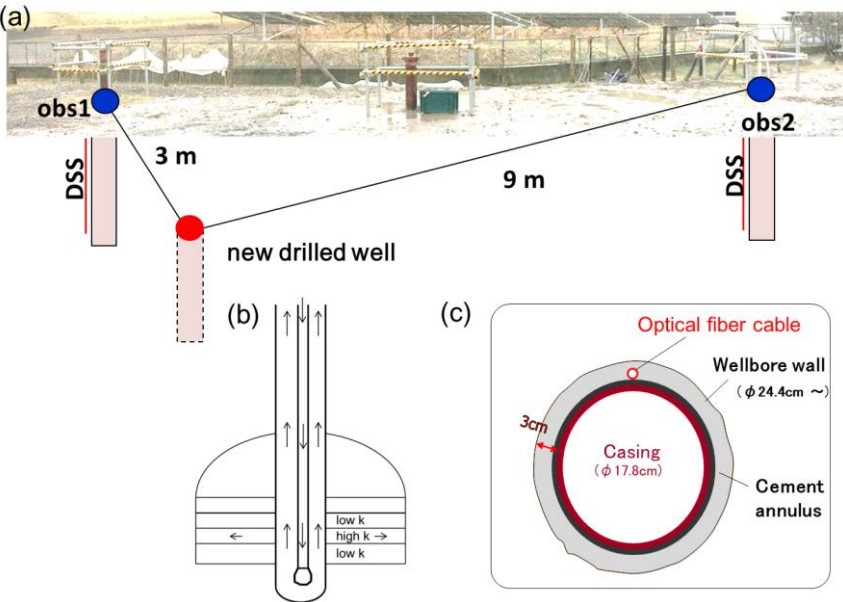

**Figure 2: (a) Well pattern for wells obs1 and obs2, in which optical fibers were installed, and the new drilled well; (b) schematic of the drilling fluid invading the reservoir formation; and (c) axial cross section of the well showing the area behind the casing installation of optical fiber cable.**

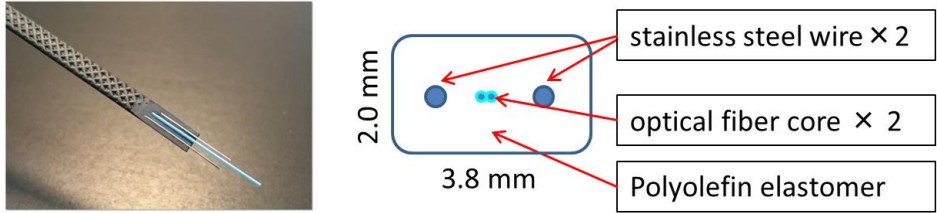

**Figure 3. Photo (left) and structure (right) of the optical fiber cable.**

Cementing operations with injection of cement slurry were undertaken to further fix the fiber cable and seal the annulus after the siting of the casing. The cementing operations must be conducted with sufficient care to ensure the integrity of the entire cementing string and avoid sudden downward migration of the cement column or the development of new local cracks or sudden compression, which, in combination with large local strains, may damage the fiber. The cable's width and height are

approximately 3.8 and 2.0 mm, respectively. Another kind of fiber cable (temperature cable) with solely sensitivity to temperature was also installed for examining the in situ temperature changes. After well completion with fiber cable installation, the wellbore and formations were equilibrated for a long duration of time (e.g., a month) to reach stability before further monitoring of reservoir testing. The data obtained during this period can be used to evaluate the cementing job and the well stability.

In this study, a new well was drilled approximately 3 m from one observation well and 9 m from the other (Fig. 2a). The diameter of the new well was approximately 15.9 cm. The final drilling depth was 186.5 m. During the drilling, a NP-700 mud pump was used to pump out and circulate the drilling fluid (mud water) flowing in the well; this was done to remove cuttings and maintain the wellbore stability. The bentonite clay-based and ribonite adjusting agents were intermittently and manually added to the drilling fluid. The drilling fluid had a density approximately 1.1 kg/L and a high viscosity (the value is unknown), which require a high pressure to drive the drilling fluid to circulate in the well.

The drilling fluid can partially invade the reservoir formation or permeable layers in the lateral direction under high pressure conditions at the wellbore (Fig. 2b). This produced hydromechanical deformation in the areas where the pressure propagated towards. The vertical changes in permeability in such lithological layers or zones are expected to guide the pattern of fluid infiltration, pressure change, and formation deformation.

We monitored the real-time strain changes at obs1 and obs2 using DSS while drilling the new well. The fiber optic acquisition was performed using the Neubrescope NBX-SR7000 device in a quick measurement mode (approximately 2 min/record). The optical fibers for the two wells were connected to the acquisition device through separate channels. We used an optical switch to routinely distribute measurement jobs to each channel.

## 4 Results and discussions

DSS records obtained during the drilling of the new well were graphed as time-depth-strain value contour images, depth-strain value profiles, and strain value-time curves (Fig. 4-6). In these figures, the time-lapse changes in strain responses accompanying the drilling process are clearly revealed at the locations of both the obs1 and obs2 wells. The spatiotemporal changes in strain are corresponding to each drilling interval. The onset of strain change corresponds to the start of the drilling process at each depth. Strain records clearly indicate the downward migration of drilling operation. Phase delays appear at both wells for strain records at depths of approximately 71, 87, and 144 m (Fig. 6a and b). The drilling process left a marked trace in the strain records (Fig. S1).

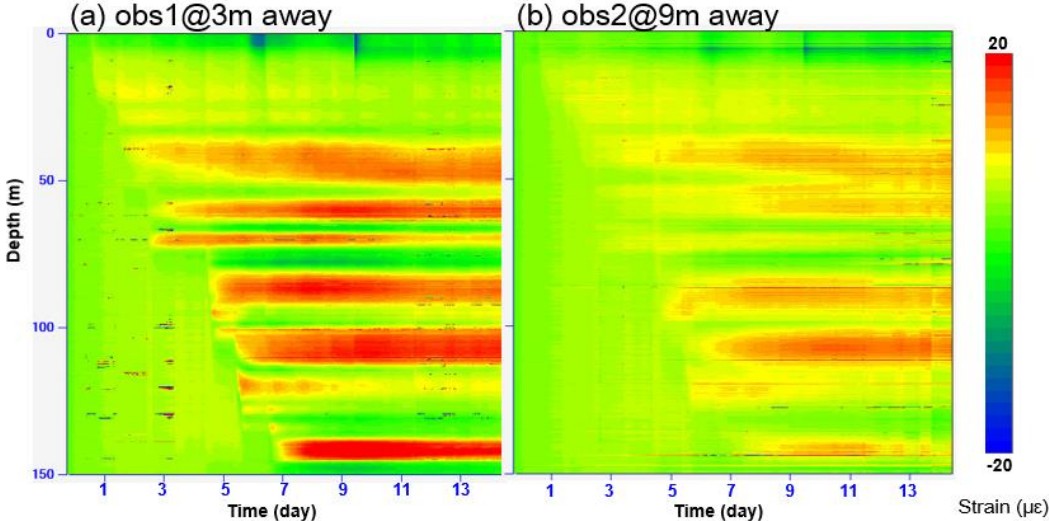

Figure 4. Strain changes with time and depth at (a) well obs1 and (b) well obs2.

Moreover, the spatiotemporal patterns of changes in strain in the two observation wells match the layered formation structure. The different magnitudes of the changes of strain in the two wells—smaller changes developed in obs2 than in obs1—may indicate the diffusion of radial pressure and attenuation from the near to far field (Fig. 4a and b) along strata. The drilling fluid invasion induced fluid pressure propagated mostly along the layers. The greatest expansion strain that developed at the closer obs1 well is approximately 25 μϵ (which is still a small value), whereas at the obs2 well it is approximately 10 μϵ (Fig. 5f). Furthermore, variations in strain magnitude in the vertical direction appear at different depths, perhaps indicating depth-dependent lithological heterogeneities (sandstone-mudstone alternation) and permeability changes. These strain peaks may indicate more permeable layers. Fig. S2 shows the well logs of compressive and shear wave velocities (Vp and Vs) in the depth range between 100 m and 150 m. The lithological changes can be also visible from Vp and Vs logs. Compared to the Vp and Vs, the distributed strain records show a clearer pattern of formation structure. In addition, there appears to be a trend in which the strain magnitude increases with respect to depth. This may be related to the increased pressure at the wall of the drilling well to greater depths, which is caused by the increasing density of drilling fluid under the effect of gravity. Among these positive strain peaks, the transition layers show negative (compressive) strains. The dilation deformation was generally larger than the compressive deformation (Fig. 5f).

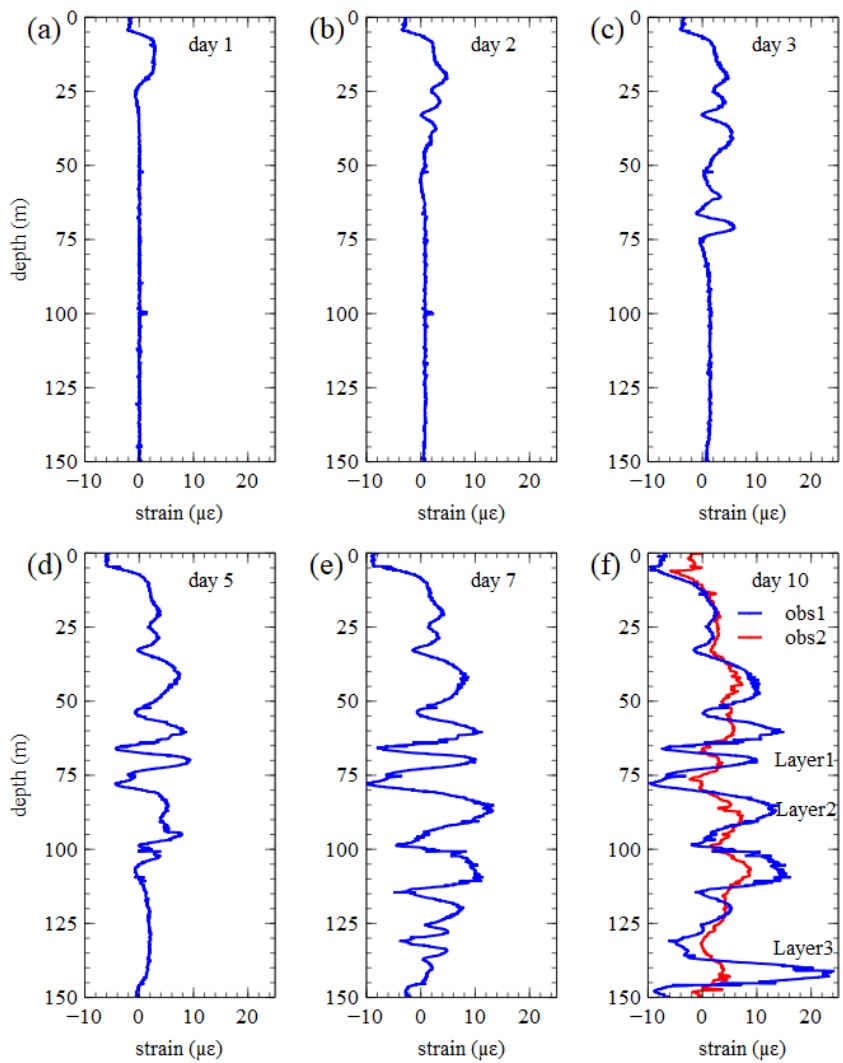

**Figure 5. Strain profiles along obs1 well on different days. The strain profile of obs2 at day 10 is added in (f) for comparison. The time series of the strain changes for the three arrows refer to depths shown in Fig. 6.**


In Fig. 6a and b, the variations in the strain values with respect to time may reflect the time-dependent pressure propagation during drilling. At the initial stage after drilling reached the depth, there were some diffusion-controlled changes as the strain increased gradually; however, after the strain developed to some values, there were some irregular variations followed by a gradual reduction in strain values. The irregular variations and reduction might be due to the instabilities of drilling operations

and the formation damage by forming of mud filter cake near the well wall during the ongoing drilling. During the drilling, water and other drilling materials were intermittently added into the drilling fluid at the surface (according to the operator's experience). Regardless, most of the raw strain data (time-series) show a quite good trend, manifesting high quality data and good DSS performance. The subtle hydromechanical deformations caused by well drilling have been clearly captured. Besides,

the changes were not relevant to temperature. The records of another optical fiber sensing cable with solely sensitivity to

temperature (and insensitive to strain) show no apparent change in temperature (Fig. S3).

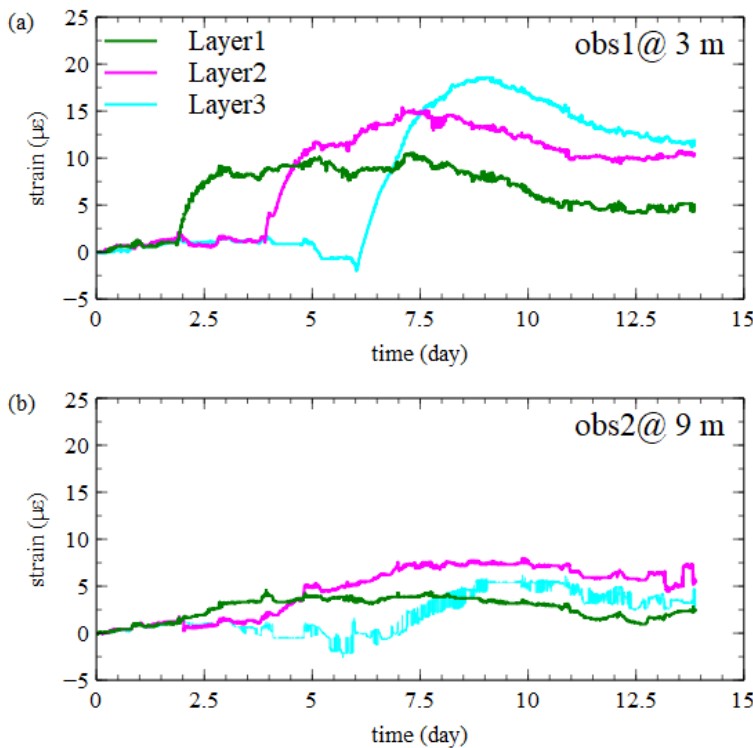

Figure 6. Strain changes with respect to time at depths approximately 71, 87, and 144 m of obs1 (a) and obs2 (b) wells.

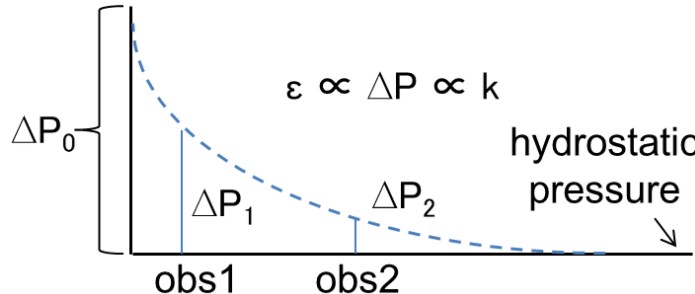

Figure 7. Schematic illustration of spatial strain ($\varepsilon$) due to changes in pore pressure ($\Delta P$). The latter is controlled by permeability
(k) of formation layers.

The strain development at obs1 and obs2 could be understood by considering the poroelastic diffusion (Biot, 1941; Rice and

Cleary, 1976; Rudnicki, 1986; Yang et al., 2015). For example, there was an additional pressure change ($\Delta P_0$) at the drilling

location due to the density increment of drilling fluid relative to the hydrostatic formation pressure. The radial pressure

diffusion caused further pressure changes ($\Delta P_1$ and $\Delta P_2$) at the depths of wells obs1 and obs2, as controlled by the permeability

of the layer (Fig. 7). Consequently, the corresponding poroelastic changes occurred for effective stress ($\sigma_1$ and $\sigma_2$) and strain ($\varepsilon_1$ and $\varepsilon_2$) at these sites.

Here we use a hydro-mechanically coupled model to simulate the poroelastic responses induced by the drilling pressure. For the drilling operation was quite dynamic (with intermittent pause and continuation events), we only consider the strain pattern
at a selected stage (which is assumed stable; day 10 in Fig. 5f). Moreover, because there are no other parameter data (such as elastic and permeability parameters) except strain records, our purpose of the modelling here is to interpret and capture the main effect of formation permeability structure on the deformation pattern but not to quantify the exact value. Essentially, we consider the extra fluid pressure in the wellbore exerted by the depth-dependent density increment of the drilling fluid and do not consider the dynamic processes (such as pause and continuation, addition of drilling mud, and pressure perturbations, etc.).
An axisymmetric cylindrical 2D model (300 m $\times$ 300 m) is built to represent the site setting. The vertical axis represents the new drilled well. We compare the modelled strain at distance of 3 m and 9 m to the vertical axis with the strain records of obs1 and obs2. The finite element modelling framework MOOSE is used to solve the coupled model (Permann et al., 2020). A Dirichlet condition with depth-dependent pressure ($= \Delta \rho g z$) is set at the drilling location and a constant pressure at the outer side. The normal component of the displacements at the outer side and bottom of the model is set to zero. We use constant
values for Young's modulus, $2.5 \times 10^8$ Pa, Poisson's ratio, 0.29, and Biot's coefficient, 1, in the entire domain. The values are rather arbitrarily selected for they are unknown. Importantly, we set distinct permeability for each layer in the hydromechanical model. We vary the permeability values to find a result with the similar strain pattern compared to the measurement.

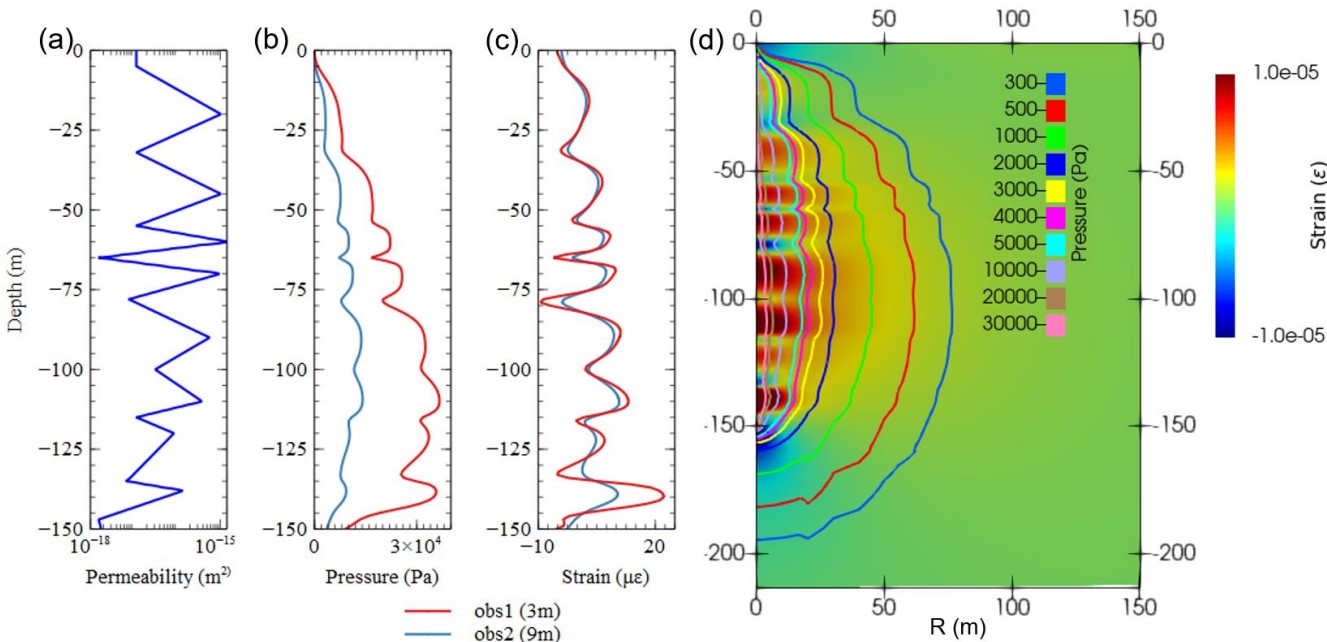


**Figure 8. Assumed permeability structure (a); profiles of the modelled pore pressure (b) and strain (c) changes at the distance of wells obs1 and obs2; and spatial image of strain with contour of pressure changes (d) on day 10.**

Fig. 8b-d show the modelling results with the assumed layered permeability structure (Fig. 8a). The modelled strain pattern on
day 10 is largely consistent with the measurement. As expected, the strain pattern reveals the main structure of the assumed permeability. This suggests that the detected strain changes are explainable by the permeability dependent poroelastic diffusion induced by the drilling. In addition, it seems that the strain records contain more abundant information of the spatial variations and are more sensitive to the formation permeability structure than the fluid pressure. The latter initially has significant variations (which are proportional to the permeability) however it gradually becomes spatially smooth in a later phase (for
example, on day 10) due to pressure diffusion.

From the modelling results, we can also observe the passive compressive deformation in the low permeability layers as in the DSS records. The compressive deformation is developed by the mechanically compacted forces exerted by the positive strain in the neighboured layers where the poroelastic expansion occurs. In the modelling, we find that the magnitude of the compressive deformation depends on the contrast of the permeability between layers (and the elastic modulus; however, not
considered here). Therefore, these low-permeability layers take a role of compensation to the positive deformation developed in those high-permeability layers although the entire formation is dominated by the dilation deformation.

In several previous studies, the surface displacement caused by fluid injection or extraction has been investigated using geodetic techniques (e.g., InSAR) and used to estimate reservoir properties (Alghamdi et al., 2020; Bohloli et al., 2018b; Bonì et al., 2020; Rezaei and Mousavi, 2019; Smith and Knight, 2019; Vasco et al., 2008). Here our results suggest that the dilation
deformation caused by fluid injection is partially compensated by adjacent zones. Therefore, using solely surface data to estimate reservoir hydraulic parameters may need to consider the compensation effect. DSS data are expected to be complementary to the surface-based monitoring methods in resolution and dimension.

The modelling is useful for examining the spatial range where has obvious pressure and strain changes. With the assumed parameters, the drilling fluid can produce a small strain (approximately 1 $\mu\epsilon$) at a distance approximately 80 m away from the
drilling well on day 10. The changes thus could be monitored by the DSS. However, we find that the spatial range, where develops clearly layered strain pattern, can be extended to approximately 30 m. Beyond the range, the layered pattern of the poroelatic strain disappears; the deformation in each layer becomes smooth and the strain magnitude becomes small. Therefore, for an observation well at a farther distance, the layered pattern could not be observed. The range is expected to be expanded with the increasing of layers' permeability and the contrast between layers and the rate of fluid injection (or extraction).


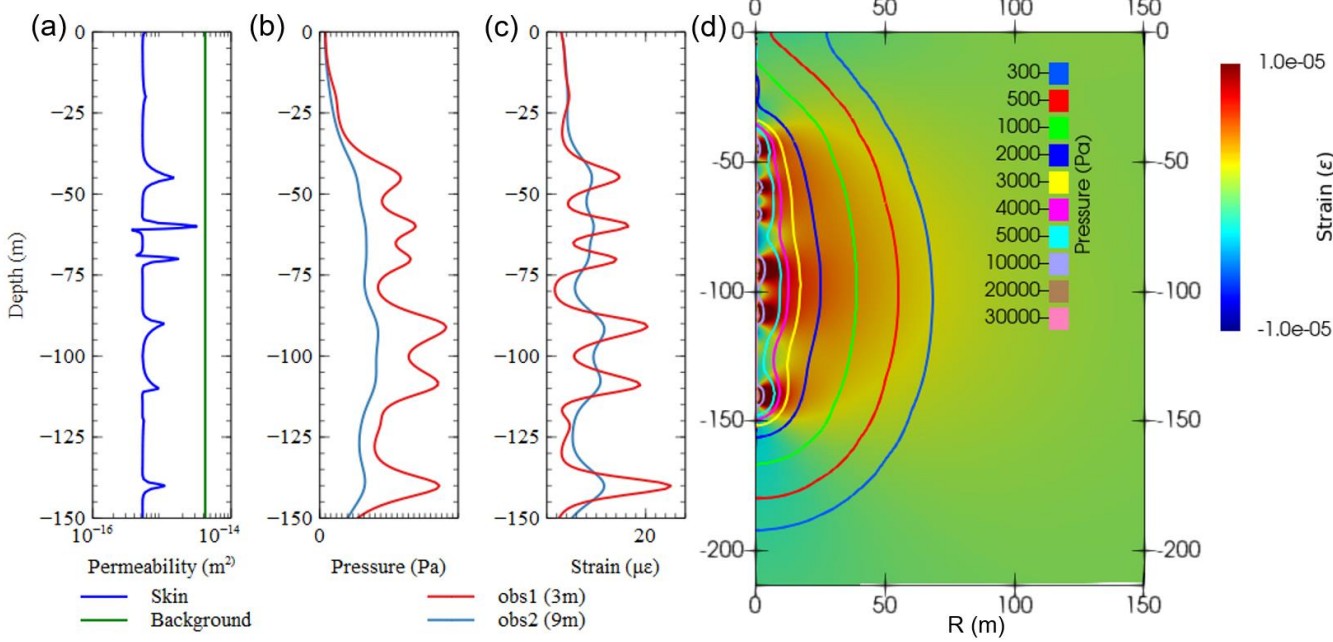

Figure 9. Assumed permeability structure of the skin formed by mud filter cakes (a); profiles of the modelled pore pressure (b) and strain (c) changes at the distance of wells obs1 and obs2; and spatial image of strain with contour of pressure changes (d) on day 10.

As above mentioned, the formation damage may be involved in the drilling process and affect the pressure diffusion process. The formation of mud filter cake near the wall of the borehole and the infiltration of solid particles in the drilling fluid may occur during the drilling and reduce the permeability around the borehole. In the above, we interpret the strain pattern is controlled by the formation's intrinsic permeability structure. Another possibility is that the formation damage and the formation of the low-permeability skin may be the source of formation heterogeneity. To investigate this possibility, here we consider a uniform formation background in permeability ($4\times10^{-15}\,\mathrm{m}^2$) and a near well skin shell (i.e., 30 cm from the wall of the drilling well) with the different degrees of permeability reduction by the mud filter cake at each depth. We make adjustment in the permeability values for each section of the skin shell to examine whether the strain pattern can be produced by heterogeneous skin. Fig. 9c-d show the modelling results with the assumed permeability values (Fig. 9a). The results suggest that the formation damage can also cause the strain pattern at obs1 and obs2. Although it is just a thin shell of mud filter cake, the resulting "strain shadow" with layered pattern can propagate to approximately 10 m away from the borehole location. Beyond the range, the fluid pressure and strain become more homogeneous. Compared to the case without formation damage, there is a larger pressure loss (with large gradient; Fig. 9d) in the nearby of the borehole and the range showing the layered pattern is narrower if the low-permeability skin is added.

As shown above, both models of layered formation with different intrinsic permeabilities and heterogeneous formation damage caused by the mud filter cakes during the drilling could result in the observed strain pattern. For uncertainties in the source (i.e., drilling) and formation parameters, we cannot rule out either of them in the data acquisition range. The real situation may include the combination of the two causes—the formation damage could be more severe for the low permeability strata. There was a chance to distinguish the two causes by conducting further investigations following the drilling, such as analysing the recovery data after the wellbore cleaning. However, the data were not recorded. Nevertheless, our modelling results suggest that the DSS records can be basically explained by the hydromechanical responses of fluid pressure diffusion.

## 5 Conclusions

Pore fluid extractions from or injections into reservoirs can induce changes in fluid pressure, modify effective stress, and deform aquifer formation. Before massive changes in mass, such fluid-to-solid hydromechanical (HM) deformations are usually subtle, linearly elastic, and recoverable; however, the deformations are often neglected because the stratum formation remains stable. In this study, we successfully measured such weak HM deformations induced by small pressure perturbations (e.g., 1 kPa) using a high-resolution DSS tool during well drilling. Both observation wells recorded the clear strain changes that accompanied well drilling operations. By numerical modelling, we have shown that the spatial pattern of deformation of the two wells may indicate the vertical permeability heterogeneity of the formation or the heterogeneous formation damage caused by the forming of mud filter cakes.

DSS provides more details of reservoir deformation along the vertical direction, which should be helpful for understanding the contribution of each layer to the overall displacement. One worthy noting issue is that the dilation deformation caused by drilling fluid injection may be compensated by adjacent layers or zones. Therefore, one may need to be cautious for the compensation effect when using solely surface geodetic data to estimate reservoir hydraulic parameters for multilayer aquifers. Vertical observation through DSS and surface-based monitoring methods (e.g., InSAR) complement each other in resolution and dimension.

This study demonstrated the good performance of a Rayleigh scattering-based DSS using TW-COTDR method. A functionality similar to the one shown here could be deployed in well testing involved with fluid injection or extraction, or in studying aquifer fluid flow behaviour with hydromechanical responses (e.g., those for natural fluids such as water, gas and oil, or those used for geological storage of $CO_2$) (Murdoch et al., 2020; Vilarrasa et al., 2013; Wu et al., 2017; Yang et al., 2019; Zappone et al., 2020). Because the high resolution and accuracy, the use of DSS would be beneficial in operations; for proper fluid injection or extraction and pressure management; the detection of fluid leakage from reservoirs (Rutqvist et al., 2016); rock fracking and stimulation (Krietsch et al., 2020),  and optimizing reservoir utilization. DSS could be also deployed in studying natural processes involving hydromechanical responses, such as at seismogenic structures (e.g., faults) related to earthquake occurrences (Guglielmi et al., 2020; Kinoshita and Saffer, 2018).

Data availability. The strain data are available at 10.6084/m9.figshare.12009504.

Author contributions. YZ participated the field work, performed the data processing and numerical analysis, and wrote the manuscript. XL gave suggestions in numerical analysis. TH and ZX contributed to project management and field work.

Competing interests. The authors declare that they have no conflict of interest.

**Acknowledgments**

This paper is based on results obtained from a project (JPNP18006) commissioned by the New Energy and Industrial Technology Development Organization (NEDO) and the Ministry of Economy, Trade and Industry (METI) of Japan.

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
