# Peer review of "In situ hydromechanical responses during well drilling recorded by fiber-optic distributed strain sensing"

_Solid Earth, 2020_

## Referee Comment (RC1) · Anonymous Referee #1 · 6 Jun 2020

This manuscript documents a quite interesting set of observations of localized deformation during shallow drilling, made with an exciting new fiber optic technology for distributed deformation sensing (based on wave scattering). That there are strains generated in the layered rock system during drilling is, I think, to be expected, but it's exciting to see this demonstrated with relatively high fidelity. I was hoping for some discussion on the frequency response at very long timescales, which would help us understand the general limitations of signal detection with DSS, but perhaps this is well beyond the scope of such a short paper.

In terms of how that deformation informs the local permeability structure, I am reluctant

to accept the results from the modeling performed here as a definitive demonstration for two main reasons:

First, the authors glance off the strong possibility of bias from an unmodeled skin effect, even though this is a known source of permeability heterogeneity; thus, they simply haven't tested whether the estimates they've obtained (or the variability between the two sampling locations) are representative of the layered system and not just related to wellbore damage and mud infiltration.

Second, it is perplexing why the authors convert the strain signals to "pressure" in order to use simplistic radial flow models. Unless the timescale of the signal is so short as to cause the system to respond like an undrained medium, strain is not simply proportional to pressure in a fully coupled poroelastic medium (not just the one way coupling they mention). This begs the question: what does this approach offer aside from introducing a whole new set of assumptions that may not hold at such a fine scale? Of course there are very simple yet powerful models of the deformation response in a poroelastic medium that could be used (e.g., Rudnicki, 1986, https://doi.org/10.1016/0167-6636(86)90042-6); using them would permit a way to model strains directly and also remark on the distribution of pore pressure changes. A more sophisticated to replicate the apparent effect of layer contrasts is also warranted.

So, overall the modeling is simply not compelling, which is a disservice to the interesting signals seen in the fiber optic records.

---

## Referee Comment (RC2) · Anonymous Referee #2 · 6 Jul 2020

Overall Comments: This manuscript presents the strain variation along two observation boreholes as a response to borehole drilling. For such a purpose, a distributed strain measurement along the two observation boreholes was conducted. The results present the effect of drilling via inducing hydromechanical deformations on the observation boreholes. Moreover, a simple hydraulic diffusion model was implemented to interpret the strain evolution in the observation boreholes. In general, this manuscript is reasonably well organized and English language errors are minor. Although the experimental part of the manuscript is innovative and nicely described especially the application of the Rayleigh spectrum for strain measurement, the numerical part of the manuscript is trivial. The authors had tried to explain the hydromechanical responses

in the observation boreholes using a simple diffusion model without considering the mechanical effect induced by drilling and rather considering only pressure propagation as the driving force for the strain variation. Overall, the reviewer considers this paper has to be extended with a hydromechanical model to describe the strain variation as well as adding more physics to the model such as skin effect. For all these reasons, I suggest this paper be accepted with major revisions and give the authors a time to consider the raised problems and enhance the scientific level of the paper. Detailed Comments: • Some authors like Kritesch et al. (2018) had used DSS for subsurface monitoring which could be addressed in L34. Here is the publication: Krietsch, Hannes, Valentin Gischig, M. R. Jalali, Joseph Doetsch, Benoît Valley, and Florian Amann. "A comparison of FBG-and Brillouin-strain sensing in the framework of a decameter-scale hydraulic stimulation experiment." In 52nd US Rock Mechanics/Geomechanics Symposium. American Rock Mechanics Association, 2018.

• It is beneficial that the authors elaborate briefly on the geology and formations of the field site. • I suggest adding the drilling progress plot to Fig. 2 and Fig. S3. • L143: I believe the authors mean Figure 2 rather than Figure 1a. • L146: I believe the authors mean Figure 2 rather than Figure S3. • Check again the cross-referencing to the figures and tables as well as citations. There are a couple of more typos. • L 173: The sentence about unstable addition of drilling fluid is not clear. Can you elaborate more on this? • To support the statement in L178, I suggest to present the temperature data in the supplementary material. • As it was mentioned above, the skin effect did not considered in the diffusion model which will affect considerably the result of the inversion model. • Moreover, the direct transformation of estimated pressure into strain in trivial.

---

## Author Comment (AC1) · 7 Aug 2020

We greatly thank the reviewers for their comments and suggestions to help improve this manuscript. Both reviewers showed interest in the monitoring results of DSS but commented on the modeling work for relating the strain changes to pore pressure and formation permeability using a hydraulic diffusion model. We first give a general response as follows.

In an earlier study (of our group), Lei et al. (2019) have shown the corresponding changes between strain and pressure signals in a pumping test in the same field (Figure 1; please see Supplement). They further performed both an analytical hydraulic

[Figure]

diffusion model, and a coupled hydromechanical model to explain the aquifer hydrome-chanical parameters, such as permeability and compressibility. Both models can give a reasonable range of permeability changes and can explain measured pressure and strain changes. The first-order strain changes were linearly related to pore pressure changes and can be interpreted using the hydraulic diffusion model. (We will elaborate this discussion in the revision.)

Therefore, in the current study, we use the hydraulic diffusion model under the first-order approximation and assume a linear relationship between strain and pressure changes with local compressibility (or storage coefficient) to consider the elastic re-sponse to pore pressure because we do not have good constraints for the other elastic constants. Moreover, the simplification with the analytical model makes it possible to match the strain or pressure curves by solving an optimization problem. Though the mechanical effect may exist, in a sense of first-order approximation, the analytical re-sults suggest that the trend and pattern of strain changes can be explainable by the hydraulic diffusion mechanism–the main physics.

Regarding the skin effect, we acknowledge that the skin effect may affect the estima-tion of permeability values (as stated in L233). Though we did aware that the impact of wellbore damage and mud infiltration when doing the analysis, the field test of using DSS monitoring during the well drilling, was the first of such a test, and these parame-ters related to skin effect, wellbore damage and mud infiltration were not independently evaluated (or not possible). In addition, during the well drilling, the well wall was self-cleaned by the drilling fluid, which was circulated from surface to bottom. Therefore, to clearly analyze the impact of the skin effect is difficult. From an analysis of the re-sponses between the two monitoring wells, we could see the skin effect (larger inverted values of permeability in obs2 than obs1) but not always. (We will explain more about this point in the revision.)

As one of the comments, we will plot both strain and pressure in revised Fig. 4 in the revision. The information may be helpful for readers to know that the hydromechanical strain (of several $\mu\varepsilon$) produced by small pressure changes (of several kPa) can be monitored by using DSS. DSS can be used not only for monitoring of mechanical deformation but also monitoring of pore pressures and fluid flow behavior. Such knowledge can be useful for designing hydraulic tests or monitoring subsurface fluid reservoirs.

Overall, with the main focus of this study is the high-resolution DSS measurement, to interpret the strain changes recorded by high-accuracy DSS during the drilling process, we try to capture the first-order factor–the diffusion of drilling-induced pressure. We acknowledge that a coupled hydromechanial model can be theoretically better; however, practically we lack further constraints besides strain records, and the drilling process maybe not ideal for such a coupled study. Another paper manuscript of us now reviewed by JGR-Solid Earth uses a fully-coupled hydromechanical model to explain the changes in strain for a well-designed and larger-scale hydraulic pumping test.

This manuscript had been previously submitted to another journal (rejected after an external review). Here I give the link of replies to the reviewers' comments (some are relevant to the comments in the current review) and take the opportunity to express my gratitude to them also.

https://docs.google.com/document/d/1HKtZK362WTT9LsQLIn4U4xjU576V1lQY2vYl-jqC_KY/edit?usp=sharing

Reference: Lei, X., Xue, Z., & Hashimoto, T. (2019). Fiber optic sensing for geomechanical monitoring:(2)-distributed strain measurements at a pumping test and geomechanical modeling of deformation of reservoir rocks. Applied Sciences, 9(3), 417.

1. This manuscript documents a quite interesting set of observations of localized deformation during shallow drilling, made with an exciting new fiber optic technology for distributed deformation sensing (based on wave scattering). That there are strains generated in the layered rock system during drilling is, I think, to be expected, but it's exciting to see this demonstrated with relatively high fidelity. I was hoping for some discussion on the frequency response at very long timescales, which would help us

understand the general limitations of signal detection with DSS, but perhaps this is well beyond the scope of such a short paper.

Re: If here I clearly understand the "frequency response at very long timescales", e.g., a long-term deformation behavior with some period (for example, seasonal), I would like to say that the monitoring of it using the DSS system is possible. Beyond this study, we have successfully tested in the same field for long-term monitoring of aquifer deformation due to seasonal agriculture water use or proposed water pumping test (e.g., about 10 days; please see Lei et al. 2019).

2. In terms of how that deformation informs the local permeability structure, I am reluctant to accept the results from the modeling performed here as a definitive demonstration for two main reasons: First, the authors glance off the strong possibility of bias from an unmodeled skin effect, even though this is a known source of permeability heterogeneity; thus, they simply haven't tested whether the estimates they've obtained (or the variability between the two sampling locations) are representative of the layered system and not just related to wellbore damage and mud infiltration. Second, it is perplexing why the authors convert the strain signals to "pressure" in order to use simplistic radial flow models. Unless the timescale of the signal is so short as to cause the system to respond like an undrained medium, strain is not simply proportional to pressure in a fully coupled poroelastic medium (not just the one way coupling they mention). This begs the question: what does this approach offer aside from introducing a whole new set of assumptions that may not hold at such a fine scale? Of course there are very simple yet powerful models of the deformation response in a poroelastic medium that could be used (e.g., Rudnicki, 1986, https://doi.org/10.1016/0167-6636(86)90042-6); using them would permit a way to model strains directly and also remark on the distribution of pore pressure changes. A more sophisticated to replicate the apparent effect of layer contrasts is also warranted.

Re: Thank you for the comments. Please see the general response. We may test the recommended Rudnicki 1986 model.

Please also note the supplement to this comment:
https://se.copernicus.org/preprints/se-2020-61/se-2020-61-AC1-supplement.pdf

**Supplement:**

**General response:**

We greatly thank the anonymous reviewers for their comments and suggestions to help improve this manuscript. Both reviewers showed interest in the monitoring results of DSS but commented on the modeling work for relating the strain changes to pore pressure and formation permeability using a hydraulic diffusion model. We first give a general response as follows.

In an earlier study (of our group), Lei et al. (2019) have shown the corresponding changes between strain and pressure signals in a pumping test at the same field (Figure 1). They further performed both an analytical hydraulic diffusion model, and a coupled hydromechanical model to explain the aquifer hydromechanical parameters, such as permeability and compressibility. Both models can give a reasonable range of permeability changes and can explain measured pressure and strain changes. The first-order strain changes were linearly related with pore pressure changes and can be interpreted using the hydraulic diffusion model. (We will elaborate this discussion in the revision.)

[Figure]

Figure 1. Observed and simulated (**a**) water head and (**b**) vertical strain (εzz) in Lei et al. (2019)

Therefore, in the current study, we use the hydraulic diffusion model under the first-order approximation, and assume a linear relationship between strain and pressure changes with local compressibility (or storage coefficient) to consider the elastic response to pore pressure because we do not have good constraints for the other elastic constants. Moreover, the simplification with the analytical model makes it possible to match the strain or pressure curves by solving an optimization problem. Though the mechanical effect may exist, in a sense of first-order approximation, the analytical results suggest that the trend and pattern of strain changes can be explainable by the hydraulic diffusion mechanism--the main physics.

Regarding the skin effect, we acknowledge that the skin effect may affect the estimation of permeability values (as stated in L233). Though we did aware that the impact of wellbore damage and mud infiltration when doing the analysis, the field test of using DSS monitoring during the well drilling, was the first of such a test, and these parameters related to skin effect, wellbore damage and mud infiltration were not independently evaluated (or not possible). In addition, during the well drilling, the well wall was self-cleaned by the drilling fluid, which was circulated from surface to bottom. Therefore, to clearly analyze the impact of the skin effect is difficult. From an analysis of the responses between the two monitoring wells, we could see the skin effect (larger inverted values of permeability in obs2 than obs1) but not always. (We will explain more about this point in the revision.)

As one of the comments, we will plot both strain and pressure in revised Fig. 4 in the revision. The information may be helpful for readers to know that the hydromechanical strain (of several με) produced by small pressure changes (of several kPa) can be monitored by using DSS. DSS can be used not only for monitoring of mechanical deformation but also monitoring of pore pressures and fluid flow behavior. Such knowledge can be useful for designing hydraulic tests or monitoring subsurface fluid reservoirs.

Overall, with the main focus of this study is the high-resolution DSS measurement, to interpret the strain changes recorded by high-accuracy DSS during the drilling process, we try to capture the first-order factor--the diffusion of drilling induced pressure. We acknowledge that a coupled hydromechanial model can be theoretically better; however, practically we lack further constraints besides strain records, and the drilling process may be not ideal for such a coupled study. Another paper manuscript of us now reviewed by JGR-Solid Earth uses a full-coupled hydromechanical model to explain the changes in strain for a well-designed  and larger-scale hydraulic pumping test.

This manuscript had been previously submitted to another journal; however, rejected by the editor after an external review. Here I give the link of replies to the reviewers' comments (some are relevant to the comments in the current review) and take the opportunity to express my gratitude to them also.

https://docs.google.com/document/d/1HKtZK362WTT9LsQLIn4U4xjU576V1lQY2vYl-jqC_KY/edit?usp=sharing

Reference:
Lei, X., Xue, Z., & Hashimoto, T. (2019). Fiber optic sensing for geomechanical monitoring:(2)-distributed strain measurements at a pumping test and geomechanical modeling of deformation of reservoir rocks. *Applied Sciences*, *9*(3), 417.

**Reviewer 1:**

1. This manuscript documents a quite interesting set of observations of localized deformation during shallow drilling, made with an exciting new fiber optic technology for distributed deformation sensing (based on wave scattering). That there are strains generated in the layered rock system during drilling is, I think, to be expected, but it's exciting to see this demonstrated with relatively high fidelity.
I was hoping for some discussion on the frequency response at very long timescales, which would help us understand the general limitations of signal detection with DSS, but perhaps this is well beyond the scope of such a short paper.

Re: If here I clearly understand the "frequency response at very long timescales", e.g., a long-term deformation behavior with some period (for example, seasonal), I would like to say that the monitoring of it using the DSS system is possible. Beyond this study, we have successfully tested in the same field for long-term monitoring of aquifer deformation due to seasonal agriculture water use or proposed water pumping test (e.g., about 10 days; please see Lei et al. 2019).

2. In terms of how that deformation informs the local permeability structure, I am reluctant to accept the results from the modeling performed here as a definitive demonstration for two main reasons:
First, the authors glance off the strong possibility of bias from an unmodeled skin effect, even though this is a known source of permeability heterogeneity; thus, they simply haven't tested whether the estimates they've obtained (or the variability between the two sampling locations) are representative of the layered system and not just related to wellbore damage and mud infiltration.

Second, it is perplexing why the authors convert the strain signals to "pressure" in order to use simplistic radial flow models. Unless the timescale of the signal is so short as to cause the system to respond like an undrained medium, strain is not simply proportional to pressure in a fully coupled poroelastic medium (not just the one way coupling they mention). This begs the question: what does this approach offer aside from introducing a whole new set of assumptions that may not hold at such a fine scale? Of course there are very simple yet powerful models of the deformation response in a poroelastic medium that could be used (e.g., Rudnicki, 1986, https://doi.org/10.1016/0167-6636(86)90042-6); using them would permit a way to model strains directly and also remark on the distribution of pore pressure changes. A more sophisticated to replicate the apparent effect of layer contrasts is also warranted.

Re: Thank you for the comments. Please see the general response. We may test the recommended Rudnicki 1986 model.

**Reviewer 2:**

1. : This manuscript presents the strain variation along two observation boreholes as a response to borehole drilling. For such a purpose, a distributed strain measurement along the two observation boreholes was conducted. The results present the effect of drilling via inducing hydromechanical deformations on the observation boreholes. Moreover, a simple hydraulic diffusion model was implemented to interpret the strain evolution in the observation boreholes. In general, this manuscript is reasonably well organized and English language errors are minor.

Although the experimental part of the manuscript is innovative and nicely described especially the application of the Rayleigh spectrum for strain measurement, the numerical part of the manuscript is trivial. The authors had tried to explain the hydromechanical responses in the observation boreholes using a simple diffusion model without considering the mechanical effect induced by drilling and rather considering only pressure propagation as the driving force for the strain variation.

Overall, the reviewer considers this paper has to be extended with a hydromechanical model to describe the strain variation as well as adding more physics to the model such as skin effect.

Re: Thank you for the comments.

We ever intuitively thought that "the mechanical effect induced by drilling" may play a role. However, after viewing the strain records, we found that, except at the very beginning of drilling to each depth (fast response to mechanical deformation acted by the drilling), the strain responses at the locations monitoring wells several meters away mainly followed a relatively slow hydraulic diffusion process. In our simple model, the latter was mainly considered to associate the strain trend and pattern to aquifer permeability structure.  Please view the general response for the modelling.

Detailed Comments:
âA˘ c Some authors like Kritesch et al. (2018) had used DSS for subsurface ´ monitoring which could be addressed in L34. Here is the publication: Krietsch, Hannes, Valentin Gischig, M. R. Jalali, Joseph Doetsch, Benoît Valley, and Florian Amann. "A comparison of FBG-and Brillouin-strain sensing in the framework of a decameter-scale hydraulic stimulation experiment." In 52nd US Rock Mechanics/Geomechanics Symposium. American Rock Mechanics Association, 2018.
âA˘ c It is beneficial that the authors elaborate briefly on the geology and formations of ´ the field site.
âA˘ c I suggest adding the drilling progress plot to Fig. 2 and Fig. S3. ´
âA˘ c L143: I believe the authors mean Figure 2 rather than Figure 1a.
â ´ A˘ c L146: I ´ believe the authors mean Figure 2 rather than Figure S3.
âA˘ c Check again the cross- ´ referencing to the figures and tables as well as citations. There are a couple of more typos.

Re: Thank you for pointing out the above problems and giving suggestions. We will address each in the revision.

âA˘ c L 173: The sentence about unstable addition of drilling fluid is not clear. Can ´ you elaborate more on this?

Re: Here "unstable addition" means the field operator did not continuously add the drilling fluid to the drilling well to cancel out the fluid loss but intermittently add by their field experience. We will revise this more clearly in the revision.

 âA˘ c To support the statement in L178, I suggest to present ´ the temperature data in the supplementary material.

Re: We will add a new figure for the data.

  c As it was mentioned above, ´ the skin effect did not considered in the diffusion model which will affect considerably the result of the inversion model.   c Moreover, the direct transformation of estimated ´ pressure into strain in trivial.

Re: Please see the discussion in the general response.

---

## Author Response (AR1)

Dear editor and reviewers:

We greatly thank the reviewers and editor for their comments and suggestions to help improve this manuscript. We have carefully addressed all the comments from two reviewers and accordingly revised the manuscript. We first give general responses as follows.

(1) Both reviewers showed interest in the monitoring results of DSS but commented on the modeling work for relating the strain changes to pore pressure and formation permeability using a hydraulic diffusion model. In the revision, we adopt the suggestions from both reviewers regarding the model and have conducted a hydromechanically coupled modeling using the FEM method.

(2) The skin effect of mud filter cakes formed during the drilling—another point commented by the reviewers—has been addressed by numerical modeling. We find that both the layered permeability structure and the heterogeneous formation of skin could possibly cause the observed strain pattern (considering the uncertainties in the source (the drilling process) and parameters).

(3) Some figures previously in the supplement have been moved to the content for better description.

The point to point replies are as follows.

Response to Reviewer 1:

1. This manuscript documents a quite interesting set of observations of localized deformation during shallow drilling, made with an exciting new fiber optic technology for distributed deformation sensing (based on wave scattering). That there are strains generated in the layered rock system during drilling is, I think, to be expected, but it's exciting to see this demonstrated with relatively high fidelity. I was hoping for some discussion on **the frequency response at very long timescales**, which would help us understand the general limitations of signal detection with DSS, but perhaps this is well beyond the scope of such a short paper.

Re: If here I clearly understand the "frequency response at very long timescales", e.g., a long-term deformation behavior with some period (for example, seasonal), I would like to say that the monitoring of it using the DSS system is possible. Beyond this study, we have successfully tested in the same field for long-term monitoring of aquifer deformation due to seasonal agriculture water use or proposed water pumping test (e.g., about 10 days; please see Lei et al. 2019). For strain sensing, the quasi static DSS method shown in this study are

through to be more suitable for monitoring of long-term behavior over DAS based strain measurement.

2. In terms of how that deformation informs the local permeability structure, I am reluctant to accept the results from the modeling performed here as a definitive demonstration for two main reasons:

First, the authors glance off **the strong possibility of bias from an unmodeled skin effect**, even though this is a known source of permeability heterogeneity; thus, they simply haven't tested whether the estimates they've obtained (or the variability between the two sampling locations) are representative of the layered system and not just related to wellbore damage and mud infiltration.

Second, it is perplexing why the authors **convert the strain signals to "pressure" in order to use simplistic radial flow models**. Unless the timescale of the signal is so short as to cause the system to respond like an undrained medium, strain is not simply proportional to pressure **in a fully coupled poroelastic medium** (not just the one way coupling they mention). This begs the question: what does this approach offer aside from introducing a whole new set of assumptions that may not hold at such a fine scale? Of course there are very simple yet powerful models of the deformation response in a poroelastic medium that could be used (e.g., Rudnicki, 1986, https://doi.org/10.1016/0167-6636(86)90042-6); using them would permit a way to model strains directly and also remark on the distribution of pore pressure changes. A more sophisticated to replicate the apparent effect of layer contrasts is also warranted.

Re: Thank you for the comments and suggestions. Please see the general response for we have added coupled numerical modeling for the layered permeability heterogeneity (considering layer contrasts) and the skin effect in the revision as reviewers' suggestions. In the new presentation, we show both the modelled strain and fluid pressure.

Response to Reviewer 2:

1. This manuscript presents the strain variation along two observation boreholes as a response to borehole drilling. For such a purpose, a distributed strain measurement along the two observation boreholes was conducted. The results present the effect of drilling via inducing hydromechanical deformations on the observation boreholes. Moreover, a simple hydraulic diffusion model was implemented to interpret the strain evolution in the observation boreholes. In general, this manuscript is reasonably well

organized and English language errors are minor.

Although the experimental part of the manuscript is innovative and nicely described especially the application of the Rayleigh spectrum for strain measurement, the numerical part of the manuscript is trivial. The authors had tried to explain the hydromechanical responses in the observation boreholes using a simple diffusion model without considering the mechanical effect induced by drilling and rather considering only pressure propagation as the driving force for the strain variation.

Overall, the reviewer considers this paper has to be extended with a hydromechanical model to describe the strain variation as well as adding more physics to the model such as skin effect.

Re: Thank you. In the revision, we have conducted a coupled hydromechanical model to model the hydromechanical responses with consideration of both the fluid pressure diffusion and mechanical effect and skin effect.

2. Some authors like Kritesch et al. (2018) had used DSS for subsurface 0 monitoring which could be addressed in L34. Here is the publication: Krietsch, Hannes, Valentin Gischig, M. R. Jalali, Joseph Doetsch, Benoît Valley, and Florian Amann. "A comparison of FBG-and Brillouin-strain sensing in the framework of a decameter-scale hydraulic stimulation experiment." In 52nd US Rock Mechanics/Geomechanics Symposium. American Rock Mechanics Association, 2018.

Re: The reference and some other relevant references have been added.

3. It is beneficial that the authors elaborate briefly on the geology and formations of 0 the field site. I suggest adding the drilling progress plot to Fig. 2 and Fig. S3.

Re: A brief introduction of the formation geology has been inserted with a reference. The drilling progress plot has been added to the strain image.

4. I believe the authors mean Figure 2 rather than Figure 1a. â 0 AËŸ c L146: I 0 believe the authors mean Figure 2 rather than Figure S3. âAËŸ c Check again the cross- 0 referencing to the figures and tables as well as citations. There are a couple of more typos.

Re: These have been addressed in the revision.

5. L 173: The sentence about unstable addition of drilling fluid is not clear. Can 0 you elaborate more on this?

Re: Here "unstable addition" means the field operator did not continuously add the drilling fluid to the drilling well to cancel out the fluid loss but intermittently add by their field experience. We have revised this more clearly in the revision.

6. To support the statement in L178, I suggest to present the temperature data in the supplementary material.

Re: A new figure demonstrating only slight temperature change has been added in the supplementary material.

7. As it was mentioned above, the skin effect did not considered in the diffusion model which will affect considerably the result of the inversion model. Moreover, the direct transformation of estimated 0 pressure into strain in trivial.

Re: Please see the general response and reply 1.

Best regards,

Yi Zhang

[revised manuscript text omitted]